# Dysregulated Retinoic Acid Signaling in the Pathogenesis of Pseudoexfoliation Syndrome

**DOI:** 10.3390/ijms23115977

**Published:** 2022-05-26

**Authors:** Matthias Zenkel, Ursula Hoja, Andreas Gießl, Daniel Berner, Bettina Hohberger, Julia M. Weller, Loretta König, Lisa Hübner, Thomas A. Ostermann, Gabriele C. Gusek-Schneider, Friedrich E. Kruse, Francesca Pasutto, Ursula Schlötzer-Schrehardt

**Affiliations:** 1Department of Ophthalmology, Friedrich-Alexander-Universität Erlangen-Nürnberg, 91054 Erlangen, Germany; matthias.zenkel@uk-erlangen.de (M.Z.); ursula.hoja@uk-erlangen.de (U.H.); andreas.giessl@uk-erlangen.de (A.G.); danielberner@gmx.net (D.B.); bettina.hohberger@uk-erlangen.de (B.H.); julia.weller@uk-erlangen.de (J.M.W.); loretta.koenig@uk-erlangen.de (L.K.); lisa.huebner@uk-erlangen.de (L.H.); thomas.ostermann@uk-erlangen.de (T.A.O.); gabriele.gusek-schneider@uk-erlangen.de (G.C.G.-S.); friedrich.kruse@uk-erlangen.de (F.E.K.); 2Genetikum, 89231 Neu-Ulm, Germany; 3Institute of Human Genetics, Friedrich-Alexander-Universität Erlangen-Nürnberg, 91054 Erlangen, Germany; francesca.pasutto@uk-erlangen.de

**Keywords:** pseudoexfoliation syndrome, pseudoexfoliation glaucoma, retinol, retinoic acid, extracellular matrix, fibrosis, TGF-β1

## Abstract

Pseudoexfoliation (PEX) syndrome, a stress-induced fibrotic matrix process, is the most common recognizable cause of open-angle glaucoma worldwide. The recent identification of PEX-associated gene variants uncovered the vitamin A metabolic pathway as a factor influencing the risk of disease. In this study, we analyzed the role of the retinoic acid (RA) signaling pathway in the PEX-associated matrix metabolism and evaluated its targeting as a potential candidate for an anti-fibrotic intervention. We provided evidence that decreased expression levels of RA pathway components and diminished RA signaling activity occur in an antagonistic crosstalk with TGF-β1/Smad signaling in ocular tissues and cells from PEX patients when compared with age-matched controls. Genetic and pharmacologic modes of RA pathway inhibition induced the expression and production of PEX-associated matrix components by disease-relevant cell culture models in vitro. Conversely, RA signaling pathway activation by natural and synthetic retinoids was able to suppress PEX-associated matrix production and formation of microfibrillar networks via antagonization of Smad-dependent TGF-β1 signaling. The findings indicate that deficient RA signaling in conjunction with hyperactivated TGF-β1/Smad signaling is a driver of PEX-associated fibrosis, and that restoration of RA signaling may be a promising strategy for anti-fibrotic intervention in patients with PEX syndrome and glaucoma.

## 1. Introduction

Pseudoexfoliation (PEX) syndrome is a common, age-related, systemic disorder of the extracellular matrix with important clinical ramifications including a particularly severe type of open-angle glaucoma and associated comorbidities affecting elastic connective tissues and the cardiovascular system [1,2]. It is associated with the excessive production and abnormal aggregation of microfibrillar components into highly cross-linked fibrillar aggregates which progressively accumulate throughout the anterior eye segment and various organ systems [3]. The typical fibrils predominantly contain elastic fiber components, such as elastin, fibrillin, fibulins, latent transforming growth factor beta binding proteins (LTBPs), microfibril-associated proteins (MFAPs), and the cross-linking enzyme lysyl oxidase-like 1 (LOXL1), and appear to be produced by various unrelated cell types including epithelial cells, endothelial cells, fibroblasts, and smooth muscle cells [4]. The pathophysiology of PEX syndrome involves a complex interplay of profibrotic factors such as TGF-β1 (transforming growth factor beta 1) and CTGF (connective tissue growth factor), proteolytic enzymes and inhibitors, pro-inflammatory cytokines, and dysregulated stress response pathways leading to its description as a stress-induced fibrosis [3].

PEX-associated glaucoma, which may account for the majority of glaucoma in some countries [1], is thought to be caused by obstruction of the aqueous humor outflow pathways with the abnormal extracellular fiber aggregates leading to intraocular pressure (IOP) elevation and pressure-induced optic nerve damage [5]. PEX glaucoma is usually associated with high IOP levels and fluctuations, rapid progression, and poor response to medical treatment, implying a severe clinical course with a high risk of blindness. Thus, any strategy to prevent or attenuate this abnormal fibrotic process could lead to a significant improvement of this serious disease. Genetic association and epidemiological studies indicate that both genetic and environmental factors contribute to the susceptibility of this complex disease. *LOXL1*, encoding a cross-linking matrix enzyme, has been recognized as the major genetic effect locus for PEX syndrome and PEX glaucoma in all populations worldwide, although the biological role of the associated variants still remains unknown [6]. Potential comodulating environmental and nutritional factors, associated with risk for PEX, include geographic latitude, ultraviolet radiation, oxidative stress, low antioxidant diet, low folate intake, high caffeine consumption, and others [7,8]. However, the precise cellular and molecular mechanisms underlying PEX syndrome and glaucoma are not yet understood and there is no specific treatment for this disease available.

Through deep sequencing of the *LOXL1* locus, we have previously identified a common non-coding sequence variant, rs7173049A > G, downstream of *LOXL1*, which was significantly associated with PEX across various populations [9]. This polymorphism represented the only known common variant at the broad *LOXL1* locus without allele effect reversal in populations of different ancestry, implying a biologically relevant effect. Functional analyses showed that this variant influenced expression levels of a neighboring gene, *STRA6* (signaling receptor and transporter of retinol STRA6)*,* a cell surface receptor regulating cellular vitamin A uptake and retinoic acid signaling [9]. STRA6 and selected components of the STRA6 receptor-driven RA signaling pathway were significantly downregulated in ocular tissues of PEX patients compared to age-matched controls. The spatial association of diminished STRA6 expression and abnormal fibrillar aggregates in tissue sections of PEX eyes suggested that an impaired vitamin A metabolism may be causally involved in the pathophysiology of PEX syndrome/glaucoma.

Vitamin A or retinol is an essential nutrient that is required for eye development and function throughout life. Diet-derived retinol is transported as a complex with retinol-binding carrier protein 4 (RBP4) and transthyretin (TTR) in the blood and taken up via the STRA6 receptor into target cells [10,11]. Intracellularly, retinol is further processed into retinaldehyde (retinal) and further into retinoic acid (RA) with the aid of binding proteins, such as CRBPs (cellular retinol binding proteins) and CRABPs (cellular retinoic acid binding proteins), and two enzyme families, the ADHs (alcohol dehydrogenases) and ALDHs (aldehyde dehydrogenases). RA, the biologically active metabolite, serves as a ligand for two subfamilies of nuclear retinoid receptors RAR (retinoic acid receptor) and RXR (retinoid X receptor) forming RAR/RXR heterodimers. The ligand-receptor complex regulates the transcriptional activity of many target genes through binding to specific DNA sequences termed retinoic acid response elements (RARE). We have previously identified the binding of the transcription factor RXRα to *LOXL1* regulatory sequences as a mechanism involved in *LOXL1* gene expression [12].

Dysregulated RA signaling has received considerable attention in the pathophysiology of fibrosis in various organ systems through processes of fibroblast activation and upregulation of matrix production, frequently via activation of TGF-β signaling [13]. Accordingly, vitamin A and its metabolites, especially all-*trans* retinoic acid (ATRA), have been shown to prevent fibrotic lesions in various experimental disease models by directly inhibiting the production of extracellular matrix and/or by inhibiting the profibrotic TGF-β pathway [13]. Based on these reports and our preliminary data, we hypothesized that RA signaling is functionally impaired in tissues of PEX patients, that this impairment and concomitant alterations in TGF-β signaling are causally involved in PEX-associated fibrosis, and that restoration of RA signaling can prevent or mitigate fibrotic alterations associated with PEX syndrome/glaucoma. Here, we performed an in-depth characterization of the alterations in RA signaling in cells, tissues, and body fluids obtained from patients with PEX syndrome and age-matched controls. We further analyzed the effects of dysregulated RA signaling on PEX-associated matrix metabolism using pharmacological and siRNA-mediated pathway inhibition in two disease-relevant cell culture models in vitro. We also investigated the crosstalk between the TGF-β and RA signaling pathways in the perpetuation of the fibrotic processes associated with PEX. Finally, we evaluated the potential anti-fibrotic effects of RA signaling pathway restoration using natural and synthetic retinoids on PEX-associated matrix metabolism in vitro.

## 2. Results

### 2.1. Alterations in Retinoic Acid Signaling in PEX Tissues In Situ

The Human Retinoic Acid Signaling RT^2^ Profiler^TM^ PCR Array was initially used to monitor the expression of 84 genes involved in RA signaling in iris specimens (n = 3) of PEX and control eyes. Genes were considered as differentially expressed when their expression levels exceeded a 1.5-fold difference in all three iris specimens analyzed. In total, 20/84 genes were consistently upregulated and 25/84 genes were downregulated in PEX specimens compared to controls (Appendix A). Significant differences were observed for *ALDH1A1* (aldehyde dehydrogenase 1 A1), *CD38*, *HSD17B2* (hydroxysteroid (17-beta) dehydrogenase 2), *PAX6* (paired box 6), *PPARD* (peroxisome proliferator-activated receptor delta), *RARB* (retinoic acid receptor beta), *SOX2* (SRY (sex determining region Y)-box 2), *TGFB2* (transforming growth factor, beta 2), and *WNT5A* (wingless-type MMTV integration site family member 5A). Genes encoding key components of the RA signaling pathway including transport, cellular uptake, and metabolism of retinol to RA as well its nuclear receptors were validated by specific qRT-PCR primer assays in a larger number of iris specimens of PEX (n = 24) and control (n = 23) eyes. Genes that were shown to be significantly downregulated in PEX specimens included, among others, *ADH1* (alcohol dehydrogenase 1), *ALDH1A1*, *CRABP1* (cellular retinoic acid binding protein 1), *CRABP2*, *PPARG* (peroxisome proliferator-activated receptor gamma), *RARA* (retinoic acid receptor alpha), *RARB*, *RBP1* (retinol binding protein 1), *RBP2*, *RBP4*, *RDH10* (retinol dehydrogenase 10), *RXRA* (retinoid X receptor alpha), *SREBF1* (sterol regulatory element binding transcription factor 1), and *STRA6*. Only one gene, *RXRG*, was confirmed to be significantly upregulated in PEX specimens (Appendix A).

To further dissect the dysregulation of RA signaling components in PEX tissues in situ, we analyzed all key components of the entire retinol pathway in tissues of both anterior eye segment (iris and ciliary body) and posterior eye segment (choroid, retina) from PEX syndrome (n = 24) and age-matched control eyes (n = 23). Real-time PCR analysis showed significant and consistent downregulation of the retinol carrier protein *RBP4*, the cellular receptor *STRA6*, the cellular retinol binding proteins *RBP1* and *RBP2*, the converting enzymes *ADH1*, *RDH10*, and *ALDH1A1*, the cellular RA binding proteins *CRABP1* and *CRABP2*, and the nuclear RA receptors *RARB*, *RXRA* and *RORC* (RAR related orphan receptor gamma) in PEX tissues compared to controls. *ALDH8A1* was only downregulated in the retina and choroid of PEX eyes. Tissue-specific dysregulation was also observed for the nuclear receptors *RARA*, *RXRB*, *RXRG*, *RORA*, and *RORB*, whereas expression levels of *RARG* showed no difference between PEX and control in any tissue type analyzed (Figure 1; Appendix A). No significant differences were also observed for *ADH2*, *ADH3*, *ADH5*, *ALDH1A2*, *ALDH1A3*, *CYP26B1* (cytochrome P450 family 26 subfamily B member 1), *LRAT* (lecithin retinol acyltransferase), *RBP3*, *RDH5*, and *TTR* (Appendix A). Expression levels of *ADH4* and the RA-inactivating enzymes *CYP26A1* and *CYP26C1* were below the limit of detection. Key components of the pathway, including *STRA6*, *RBP1*, *ADH1*, *ALDH1A1*, and *RARA*, were consistently reduced in all disease stages including early stages of PEX prior to the appearance of distinct clinical signs, suggesting a fundamental role of disturbed RA signaling in pathophysiology (Appendix A). No sex-specific differences in expression levels of RA pathway components were found.

Reduced protein expression levels of STRA6, RBP1, and RBP4 had been already previously demonstrated in the iris and ciliary body tissues, aqueous humor, and serum samples of PEX patients compared to age-matched controls [9]. Therefore, this study focused on the protein expression of the major nuclear receptors RARα and RXRα in PEX and control tissues to determine PEX-associated alterations in RA signaling on the cell and tissue level. RARα and RXRα could be immunolocalized to most cell nuclei in normal ocular tissues, including the iris and ciliary body, indicating active RA signaling in these cells (Figure 2A,B). Nuclear staining for RARα was more prominent than that for RXRα. In corresponding tissues of PEX eyes, markedly reduced expression levels of RARα and RXRα were observed, often in association with LOXL1-positive PEX material deposits, indicating that RA signaling is reduced in PEX cells. In contrast, immunolabeling of the TGF-β signaling mediators and transcription factors Smad2/3 revealed a marked nuclear expression pattern, in close relation with PEX material deposits, in PEX tissues, whereas only weak nuclear staining was observed in corresponding control tissues (Figure 2C). This observation suggests activation of TGF-β mediated Smad signaling in PEX cells and tissues.

Significantly reduced expression levels of RARα, indicated by specific bands at 51 kDa, were also confirmed by Western blotting in PEX tissue specimens compared to controls (Figure 3A). Electrophoretic mobility shift assays (EMSAs) using oligonucleotides containing RAR and RXR consensus binding sequences (RARE) and nuclear protein extracts from human Tenon’s capsule fibroblasts (hTCF) of PEX and control patients showed specific DNA-protein complexes (Figure 3B). Shifted bands could be completely inhibited by a 200-fold molar excess of unlabeled oligonucleotides (competitor), proving specificity of binding. Quantitative analysis of protein-DNA complexes revealed significant differences with reduced binding of PEX nuclear extracts compared to control extracts, corroborating diminished RA signaling in cells of PEX patients. To further assess the potential effect of activated TGF-β signaling on the RA signaling pathway, hTCF and human trabecular meshwork cells (hTMC) were stimulated with TGF-β1. Compared to untreated control cells, TGF-β1 significantly downregulated the expression of *RARA* and *RXRA* in both cell types, indicating crosstalk between both pathways. Treatment of both cell types with the TGF-β1 inhibitor SB 431542 reversed the inhibitory effect, proving the specificity of RAR and RXR regulation by TGF-β1 (Figure 3C).

To further evaluate potential alterations in vitamin A uptake and transport, levels of retinol carrier proteins RBP3, RBP4, and TTR were analyzed in aqueous humor (n ≥ 15) and serum samples (n = 12) of PEX and control patients using enzyme-linked immunosorbent assays (ELISA). Aqueous humor levels of RBP3 were significantly reduced from 5.26 ± 2.0 ng/mL in cataract patients to 4.05 ± 1.0 ng/mL (*p* = 0.009) in PEX patients; levels of RBP4 were reduced from 93.07 ± 61.95 ng/mL to 45.17 ± 22.27 ng/mL (*p* = 0.02), and levels of TTR were reduced from 10.11 ± 5.45 µg/mL to 5.68 ± 1.19 µg/mL (*p* = 0.01) (Figure 4A). In contrast, serum levels of all carrier proteins were not different between controls (RBP3: 0.66 ± 0.22 ng/mL; RBP4: 251.0 ± 131.2 µg/mL; TTR: 2.89 ± 1.14 mg/mL) and PEX patients (RBP3: 0.66 ± 0.27 ng/mL; RBP4: 208.40 ± 97.24 µg/mL; TTR: 2.71 ± 1.65 mg/mL) (Figure 4A). In cataract samples, aqueous humor levels of RBP3, RBP4, and TTR correlated significantly with total aqueous protein levels (Appendix A). In contrast, PEX samples showed a significant correlation between TTR and total protein levels only, whereas no correlation was observed for RBP3 and RBP4 levels, indicating independence of aqueous RBP values from the blood-aqueous barrier function.

Serum retinol levels, measured by high-performance liquid chromatography (HPLC), in PEX patients (532.3 ± 142.4 ng/mL; n = 34) and those of control patients (489.0 ± 68.9 ng/mL; n = 34) were not significantly different, suggesting a local dysregulation of RA signaling confined to the ocular compartment (Figure 4B). However, in both groups, serum retinol levels were significantly higher in males (control: 523.1 ± 63.3 ng/mL; PEX: 602.2 ± 136.2 ng/mL) than in females (control: 470.4 ± 64.6 ng/mL; PEX: 483.4 ± 125.3 ng/mL), which is in agreement with published studies [14].

### 2.2. Involvement of Dysregulated Retinoic Acid Signaling in PEX-Associated Fibrosis In Vitro

To analyze the downstream effects of dysregulated RA signaling on matrix metabolism, we used two approaches to block RA signaling in vitro, i.e., gene silencing and pharmacologic inhibition of RA nuclear receptors, in primary human cell types, hTCF and hTMC, which are known to be involved in PEX pathogenesis. These cell types are known to secrete and assemble an abundant extracellular matrix including an extensive elastic fiber network in vitro. siRNA-mediated knockdown of nuclear receptors resulted in suppression of *RARA* and *RXRA* expression down to 10% (Figure 5A). Efficient knockdown was also confirmed on the protein level for RARα after 48 h in both cell types (Figure 5B). *RARA* and *RXRA* silencing resulted in a consistent and significant upregulation of a broad spectrum of PEX-relevant genes, including *LOXL1*, *ELN* (elastin), *FBN1* (fibrillin-1), *LTBP2*, *FBLN5* (fibulin-5), *MFAP2* and *TGFB1* as well as general pro-fibrotic genes, including *COL1A1* (collagen type I alpha 1), *COL3A1* (collagen type III alpha 1), *COL4A1* (collagen type IV alpha 1), and *FN1* (fibronectin-1), in both hTCF and hTMC compared to control cells transfected with scrambled (non-targeting) siRNA (Figure 5A). Upregulation of *LTBP1*, *MMP2* (matrix metalloproteinase-2), *TIMP2* (tissue inhibitor of metalloproteinases 2), and *TGFBR2* (transforming growth factor beta receptor 2) occurred in a cell-type or receptor-specific pattern only. The significant upregulation of *TGFB1* and its downstream mediator *CTGF* in both RA-deficient cell types raised the possibility that RA deficiency induces activation of TGF-β signaling. Expression levels of *FBLN4* were not affected by *RARA* or *RXRA* silencing. On the protein level, knockdown of *RARA* (Figure 5C) led to a significant upregulation of LOXL1, elastin, and fibrillin-1 in both hTCF and hTMC compared to scrambled siRNA-transfected control cells.

In addition, pharmacological RA pathway inhibitors, including the aldehyde dehydrogenase inhibitors disulfiram and liarozole, the pan RAR-antagonists AGN 193109 and BMS 493, and the pan-RXR-antagonists HX 531 and UVI 3003, were tested at various concentrations on expression levels of signature genes. Only BMS 493 at a concentration of 5 µM significantly affected gene expression and consistently upregulated levels of *LOXL1*, *ELN*, *FBN1*, *LTBP1* and *LTBP2*, *FBLN4* and *FBLN5*, *TGFBR2*, *COL1A1*, *COL3A1*, *COL4A1*, and *FN1* in both hTCF and hTMC compared to untreated control cells (Figure 6). Expression levels of *MFAP2*, *TGFB1*, *CTGF*, *MMP2*, and *TIMP2* were either not affected or in one cell type only by this pharmacological approach.

### 2.3. Effects of Retinoic Acid Signaling Activation on PEX-Associated Fibrosis In Vitro

To analyze the downstream effects of activated RA signaling on matrix metabolism in vitro, we stimulated RA signaling by administration of the natural retinoid ATRA in hTCF and hTMC. Administration of ATRA was preferred to 9-*cis* RA, which has a much more limited presence in adult tissues than ATRA [13]. In initial dose-response experiments, ATRA was found to significantly reduce expression levels of signature genes *LOXL1*, *ELN*, and *FBN1* in hTCF at a concentration of 1–10 µM (Appendix A). At the chosen concentration of 2 µM, ATRA significantly upregulated mRNA expression levels of *RARA* and *MMP2*, but significantly downregulated levels of PEX-relevant genes *LOXL1*, *ELN*, *FBN1*, *LTBP1*, *LTBP2*, *FBLN4*, *FBLN5*, *MAGP1*, and *TIMP2*, and pro-fibrotic genes *COL1A1*, *COL3A1*, *COL4A1*, and *FN1* in both hTCF and hTMC compared to untreated control cells (Figure 7A). No effect was seen on the expression of *TGFB1*, *TGFBR2*, and *CTGF*. As shown for selected candidate genes *LOXL1*, *ELN*, *FBLN1*, and *LTBP1*, which represent major constituents of PEX material, ATRA also downregulated expression levels in other PEX-relevant epithelial cell types, i.e., non-pigmentary ciliary epithelial cells, iris pigment epithelial cells, and lens epithelial cells (Figure 7B).

In addition to the natural retinoids ATRA and 9-*cis* RA, which act as high-affinity ligands for all isoforms of RAR and RXR [13], the efficiency of isoform-specific synthetic retinoids on the modulation of matrix expression were additionally explored in hTCF [15]. Selective RARα (AM 80), RARβ (AC 261066), RARγ (CD 1530), RXRα (CD 3254), and pan-RXR (LGD 1069; bexarotene) agonists were used at concentrations of 0.01–10 µM (Appendix A). Compared to untreated control cells, AM 80, AC 261066, CD 3254, and LGD 1069 were found to significantly reduce expression levels of candidate genes *LOXL1*, *ELN*, *FBN1*, and *LTBP1* at 10 µM with similar efficiency as 2 µM ATRA or 2 µM 9-*cis* RA (Figure 7C). The inhibitory effect of the RARγ agonist CD 1530 was confined to *LOXL1* and *LTBP1* gene expression only. These data are consistent with altered expression levels of RARα, RARβ, RXRα, RXRβ, and RXRγ, but lack of differences in expression levels of RARγ in tissues of PEX patients.

Comparative analyses of the effects of RA pathway inhibition by BMS 493 and RA pathway activation by ATRA on gene expression levels at baseline or in response to TGF-β1 treatment confirmed significant suppression of basal mRNA levels of *LOXL1*, *ELN*, *FBN1*, and *LTBP1* by the RAR agonist ATRA and induction of mRNA levels by the RAR antagonist BMS 493 in both hTCF and hTMC (Figure 8A). ATRA was also found to downregulate the TGF-β1-induced expression levels of these genes. This suppressive effect was even more prominent on the protein level in both hTCF and hTMC, where ATRA significantly downregulated TGF-β1-induced levels of LOXL1, elastin, and fibrillin-1 via suppression of phosphorylation of Smad2 (Figure 8B). These findings indicate that the matrix-modulating effects of ATRA are mediated by the inhibition of TGF-β1 signaling.

Immunofluorescence analysis of the microfibrillar network deposited by hTCF in vitro showed an increase in fibrillin-1 staining upon treatment with RA pathway inhibitor BMS 493, similar to that induced by TGF-β1, but with a marked decrease upon treatment with ATRA, the pan-RXR agonist LGD 1069 (Figure 9A), the RARα agonist AM 80, and the RARβ agonist AC 261066 (not shown). Quantitative analysis of fluorescence intensity of fibrillin-1 staining across whole images (n = 20 per group) confirmed a significant increase in staining intensity after treatment with BMS 493 and a significant reduction after treatment with ATRA and synthetic retinoids compared to untreated control cells (Figure 9B). ATRA was also able to significantly suppress the TGF-β1-induced matrix accumulation. A pronounced suppression of TGF-β1-induced production of microfibrils by hTCF after ATRA treatment could also be observed on the ultrastructural level in 3D culture systems (Figure 9C). Double labeling of elastin, fibrillin-1, and LTBP-1 with LOXL1 showed a marked reduction in positively-labeled extracellular fibrils deposited by hTCF following treatment with ATRA and LGD 1069 for 7 days compared to untreated and TGF-β1 treated control cells (Figure 10). TGF-β1-induced cytoplasmic expression of LOXL1 and F-actin, as indicated by staining with phalloidin, appeared to be also reduced by ATRA. Moreover, ATRA markedly decreased TGF-β1-induced nuclear translocation of Smad2/3, indicating suppression of TGF-β1 signaling (Figure 10).

## 3. Discussion

Several lines of evidence pointed to an involvement of the RA signaling pathway in PEX pathogenesis. Previous studies showed a significant and consistent association between PEX syndrome and a common non-coding variant, rs7173049A > G, downstream of *LOXL1* throughout all populations, which represented a plausible candidate for a true causal relationship with the PEX phenotype [9]. This unique variant influenced expression levels of *STRA6*, which mediates the rate-limiting transport of retinol into target cells [10,11,16], leading to significantly reduced mRNA and protein levels of this vitamin A receptor in ocular tissues of PEX patients compared to age-matched controls. Dysregulated expression levels of STRA6 in PEX tissues were further suggested to be influenced by a newly identified synonymous coding variant at *STRA6*, which was significantly associated with PEX on a genome-wide level [9]. Moreover, *LOXL1*, the major susceptibility gene for PEX syndrome and glaucoma, was recently identified as a target for RXRα binding to regulate its transcription and expression [12].

Based on this previous evidence, we performed a comprehensive experimental evaluation of the role of the RA signaling pathway in PEX-associated matrix metabolism, analyzed its crosstalk with the profibrotic TGF-β1 pathway, and evaluated its targeting as a potential anti-fibrotic intervention in PEX-associated fibrosis. We showed significant alterations in aqueous humor levels of retinol transport proteins and tissue expression levels of RA pathway components, particularly the nuclear receptors RARα and RXRα, resulting in a diminished RA signaling activity in ocular cells and tissues of PEX patients compared with age-matched controls. Our observation corresponds to a previous proteomic approach showing that RBP3 was among the most highly downregulated proteins in the aqueous humor of PEX patients [17]. RA pathway alterations were already observed in the early stages of PEX disease, supporting the notion that local dysregulation of RA signaling may be an initiating event in the development of PEX syndrome. Genetically governed downregulation of STRA6 receptor-driven RA signaling appears to be amplified by hyperactivated TGF-β1/Smad signaling, as reflected by prominent nuclear translocation of Smad2/3 in PEX tissues. In accordance with this notion, TGF-β1 treatment of hTCF and hTMC was shown to antagonize RA signaling by downregulating the expression of RARα and RXRα, whereas inhibition of TGF-β1 reverted these alterations.

Using disease-relevant cell culture models, gene silencing and pharmacologic modes of RA pathway inhibition induced the expression of profibrotic genes and the production of extracellular matrix components, which constitute major constituents of fibrillar PEX aggregates, along with an upregulation of TGF-β1, TGF-β receptor 2, and CTGF. Conversely, RA signaling pathway restoration by administration of natural and synthetic retinoids was able to suppress PEX-associated matrix production and formation of microfibrillar networks via antagonization of Smad-dependent TGF-β1 signaling. This suppressive effect of ATRA could be observed in various ocular cell types, including fibroblasts, trabecular endothelial cells, ciliary epithelial cells, lens epithelial cells, and iris pigment epithelial cells, which are all known to be involved in PEX material production [1].

By contrast, serum levels of retinol and retinol transport proteins were within the normal range when compared with age-matched control samples. It is, therefore, assumed that the pathologic changes in PEX tissues are influenced by reduced local availability of RA, independent of systemic vitamin A availability. Interestingly, STRA6 is mandatory for ocular vitamin A uptake, whereas vitamin A uptake in peripheral tissues can be maintained by STRA6-independent mechanisms when sufficient amounts of vitamin A are available [18]. Local deficiency of STRA6 receptor-driven RA signaling and an excess of activated TGF-β1, entering the eye through a defective blood-aqueous barrier [19], may create a permissive environment for the induction of fibrotic processes. The antagonistic interaction between RA and TGF-β1 signaling may fuel a vicious cycle, perpetuating the abnormal matrix production and accumulation in ocular tissues of PEX patients. Altogether, the present findings indicate that deficient RA signaling, in conjunction with hyperactivated TGF-β1/Smad signaling, is a driver of PEX-associated fibrosis and that restored RA signaling attenuates fibrotic processes through suppression of TGF-β/Smad signaling. This notion is supported by studies showing, for example, that the silencing of *STRA6* can cause tissue fibrosis via suppression of STRA6 cascades and upregulation and activation of TGF-β1 [20].

In view of the known environmental risk factors for PEX development [7,8], deficient retinoid metabolism in PEX tissues may be influenced not only by genetic variation but also by environmental and dietary factors, further augmented by a physiological reduction in retinoid levels with age [21]. In particular, UV radiation, which has been associated with an increased risk of PEX, has long been known to suppress mRNA and protein expression of retinoid receptors RAR and RXR, creating a functional vitamin A deficiency and photoaging of the human skin [22]. Consistently, topical RA treatment reduced RA receptor loss, increased RA signaling, and mitigated the damaging effects of UV radiation on human skin.

As a nutritional vitamin not produced within the body, retinoids must be ingested from animal origin as retinol or from plants as retinyl esters [10]. Frequent consumption of green and yellow vegetables and fruits, which contain provitamin A carotenoids as dietary precursors of retinol, was found to correlate with a decreased risk of PEX [23]. Traditional Inuit food includes fish and liver, which are rich in biologically active preformed vitamin A. It is, therefore, tempting to speculate that the absence of PEX syndrome among Inuits [24] may be associated with their increased consumption of a traditional vitamin A-rich diet [25]. Along these lines, a considerable percentage (23.3%) of patients with an inherited syndrome manifesting with corneal snowflake dystrophy, oculocutaneous pigmentary disturbances, and PEX syndrome at advanced age also suffered from vitamin A malabsorption [26].

Further evidence for a central role of the vitamin A and RA signaling pathways in PEX development is provided by anatomical studies. The RA signaling pathway exerts multiple and vital functions which are essential for eye development and homeostasis throughout life [27]. Evidence is accumulating to suggest that RA may be also an important molecular signal in the postnatal control of eye size [28]. *STRA6* mutations causing deficient RA signaling can contribute to a spectrum of developmental abnormalities, including micro/anophthalmia, heart defects, pulmonary dysplasia, and diaphragmatic hernia [29]. A similar disease spectrum, including relative anterior microphthalmos, atrial fibrillation, obstructive pulmonary disease, and inguinal hernias, has been reported to occur more frequently in PEX patients [2,30]. Interestingly, a higher axial length of the globe was significantly associated with a decreased risk of 12-year incidence of PEX in the Thessaloniki Eye Study [31]. Based on their observations, the authors suggested that shorter axial length, which depends on ocular vitamin A availability, is a contributing factor to the development of PEX.

In general, impaired dietary uptake, transport, and conversion of vitamin A into RA, followed by dysregulated RA signaling, has been found to be associated with many developmental and age-related ocular diseases, including glaucoma. Significantly lower levels of TTR and RBP3 were measured in the aqueous humor of primary congenital glaucoma patients [32]. In the Rotterdam Eye Study, a high intake of retinol equivalents was reported to have a protective effect on open-angle glaucoma with a two-fold lower risk of glaucoma compared to those with a low intake [33]. Systemic conditions associated with deficient RA signaling include neurodegenerative and fibrotic disorders. Disruption of RA signaling has been observed in the brain of Alzheimer‘s patients and causes deposition of amyloid beta in the brain of animal models, which could be attenuated by the administration of RA [21,34,35]. However, dysregulated RA signaling has received particular attention in the pathophysiology of fibrosis in various organ systems through processes of fibroblast activation and increased production of matrix components, such as collagen, laminin, fibronectin, and elastin [36]. It has been also shown that RA deficiency induces excessive activation of TGF-β signaling which may be responsible for excessive matrix production [37]. In the majority of experimental studies, these alterations in abnormal matrix production and hyperactivated TGF-β signaling could be attenuated or even reversed by ATRA treatment, indicating an effective therapeutic utility of retinoids for anti-fibrotic interventions [13,38,39]. Thus, ATRA treatment was able to (i) reduce the expression of collagen types I, III, and IV, fibronectin, laminin, elastin, and proteoglycans; (ii) downregulate the expression of TGF-β1, CTGF, and TGF-β receptor 2; (iii) suppress TGF-β1-induced Smad2/3 phosphorylation; and (iv) increase the expression of MMPs, such as MMP-2 and MMP-3, in various fibrosis models [39,40,41,42,43,44]. From these studies, it was concluded that ATRA can prevent organ fibrosis and that the anti-fibrotic effects are mainly mediated by the downregulation of TGF-β1 signaling and upregulation of MMP activity. These findings are largely consistent with the observations of the present study, showing downregulation of PEX-relevant matrix expression and deposition, suppression of TGF-β1-induced Smad2 phosphorylation, and upregulation of MMP-2 by ATRA, substantiating its potential for anti-fibrotic intervention in PEX-associated fibrosis. Apart from inhibiting matrix production and TGF-β signaling, ATRA was also shown to protect against oxidative stress and hypoxia-induced damage [45,46], reduce the expression of pro-inflammatory cytokines such as IL-6 and IL-17A [47,48], and reduce the expression of apolipoprotein E, a major component of fibrillar PEX aggregates [49], in various cells and tissues [50]. Since chronic stress conditions, such as oxidative stress, ischemia/hypoxia, and low-grade inflammatory processes, constitute major mechanisms involved in triggering the pathologic tissue fibrosis in PEX syndrome [3], ATRA administration might exert multiple beneficial effects in PEX pathogenesis.

Similar effects to those of ATRA were also achieved by using receptor-selective synthetic retinoids, such as the RARα agonist AM580 and the RARγ agonist R667, which were found to be effective for the suppression of subretinal fibrosis associated with proliferative retinal disease [51], suppression of retinal fibrosis [52], or the inhibition of scar formation after glaucoma filtering surgery [53]. Increased collagen levels in myocardial fibrosis in diabetic rats were reduced by treatment with the RXR agonist bexarotene [54]. By acting on specific RA receptor isoforms, synthetic retinoids have superior pharmacokinetic properties and fewer side effects than naturally occurring retinoids. Thus, retinoids are currently being evaluated in many therapeutic areas for various diseases including cancer, neurodegenerative diseases such as Alzheimer’s and Parkinson’s disease, diabetes, organ fibrosis, and skin disorders such as psoriasis, sclerosis, and photoaging [55].

In conclusion, the findings of this study provided evidence of a link between PEX-associated fibrosis and impaired RA signaling in ocular tissues which may be influenced by genetic and environmental factors. Dysregulated RA signaling has the potential to contribute to PEX pathogenesis through multiple downstream pathways which are known to be involved in PEX pathogenesis, including TGF-β1 and Wnt signaling pathways, autophagy, and mitochondrial dysfunction [3,56]. The interaction of the RA signaling pathway with all these pathways is well documented [57,58,59]. Thus, the RA signaling pathway could reasonably be targeted to prevent or treat fibrotic alterations in the eye caused by insufficient local retinoid levels in PEX patients. However, any potential application of ATRA or receptor-selective retinoids for the treatment of PEX-associated fibrosis requires future studies, e.g., using specific cell culture models that recapitulate the abnormal matrix metabolism of PEX. Potential challenges may arise from the variable effects of retinoids on different cell types, receptor isoform-specific effects, and diverse dose-dependent effects [13,36,60].

## 4. Materials and Methods

### 4.1. Human Tissues and Study Approval

Human donor eyes used for corneal transplantation with appropriate research consent were obtained from donors of European origin and processed within 20 h after death. Informed consent to tissue donation was obtained from the donors or their relatives, and the protocol of the study was approved by the Ethics Committee of the Medical Faculty of the Friedrich-Alexander-Universität Erlangen-Nürnberg (No. 4218-CH) and adhered to the tenets of the Declaration of Helsinki for experiments involving human tissues and samples. The use of aqueous humor and serum samples with appropriate research consent was approved by the Ethics Committee of the Medical Faculty of the Friedrich-Alexander-Universität Erlangen-Nürnberg (No. 138_18B). Written informed consent was obtained from all individuals.

For RNA extractions, 24 donor eyes with PEX syndrome (mean age, 74.9 ± 7.9 years; 11 male, 13 female) and 23 normal-appearing age-matched control eyes without any known ocular disease (mean age, 76.0 ± 9.2 years; 12 male 11 female) were used. Patients were classified as early (n = 7) or late (n = 17) stage PEX syndrome according to the amount of macroscopically visible PEX material deposits on anterior segment structures and subsequently confirmed by electron microscopic analysis of small tissue sectors, as described previously [61]. For protein extractions, three donor eyes with manifest PEX syndrome (mean age, 72.0 ± 5.7 years; 1 male, 2 female) and three normal-appearing age-matched control eyes without any known ocular disease (mean age, 76.7 ± 8.2 years; 1 male, 2 female) were used. For immunohistochemistry, ocular tissues were obtained from five donor eyes with PEX syndrome (mean age, 74.2 ± 5.8 years) and five control eyes (mean age, 75.6 ± 9.8 years). Ocular tissues were prepared under a dissecting microscope and shock frozen in liquid nitrogen.

Aqueous humor was aspirated during cataract surgery from 50 patients with PEX syndrome (mean age, 78.3 ± 6.0 years) and 50 patients with cataract only (mean age, 74.3 ± 7.8 years). Serum samples were collected from 15 patients with PEX syndrome (mean age, 72.6 ± 10.3 years) and 15 healthy subjects (mean age, 68.7 ± 12.3 years). All samples were immediately frozen in liquid nitrogen and stored at −80 °C. For analysis of serum vitamin A levels, fasting peripheral venous blood samples were obtained from 34 patients with PEX syndrome (mean age, 74.3 ± 9.5 years; 14 male, 20 female) and from 34 patients with Fuchs endothelial corneal dystrophy (FECD) as the control group (mean age, 68.6 ± 10.2 years; 12 male, 22 female). Blood samples were protected from light and immediately sent to the Central Laboratory of the Universitätsklinikum Erlangen for high-performance liquid chromatography according to the standard method. All individuals underwent routine ophthalmologic examination for clinical signs of PEX syndrome after pupillary dilation. Patients with a history of vitamin A deficiency, vitamin A supplementation, diabetes, liver disease, infectious disease, autoimmune disease, and significant eye disorders (including retinal disorders) other than cataract, PEX syndrome, or FECD were excluded from the study.

### 4.2. Real-Time RT-PCR

Ocular tissues, homogenized with the Precellys 24 homogenizer and lysing kit (Bertin; Frankfurt, Germany), and cultured cells were extracted using the RNeasy Mini kit (Qiagen, Hilden, Germany) including an on-column DNase I digestion step. Differential gene expression analysis was performed with the Human Retinoic Acid Signaling RT^2^ Profiler PCR array (Qiagen; Hilden, Germany) profiling the expression of 84 key genes involved in retinoic acid signaling (Appendix A) using 75 ng of total RNA preamplified with the Preamp cDNA Synthesis kit (Qiagen; Hilden, Germany) and the RT^2^ SYBR Green qPCR master mix (Qiagen; Hilden, Germany) according to the manufacturer’s protocol. Data were analyzed using the RT^2^ Profiler PCR array data analysis tool version 3.2 (Qiagen; Hilden, Germany).

Array results were partly confirmed using specific qRT-PCR primer assays. First-strand cDNA synthesis was performed using 0.5 µg of total RNA and Superscript II reverse transcriptase (Invitrogen; Karlsruhe, Germany) as previously described [61]. PCR reactions were performed using the CFX Connect thermal cycler and software (Bio-Rad Laboratories, München, Germany). PCR reactions were run in duplicate and contained diluted first-strand cDNA, 0.4 µM each of upstream- and downstream-primer, and 1x SsoFast EvaGreen Supermix (Bio-Rad Laboratories, München, Germany), according to the manufacturers’ recommendations. Primer sequences (Eurofins; Anzing, Germany) designed using Primer 3 software and reaction conditions are given in Appendix A. For normalization of gene expression levels, ratios relative to the housekeeping genes *GAPDH* (glyceraldehyde-3-phosphate dehydrogenase) and *HPRT1* (hypoxanthine phosphoribosyltransferase 1) were calculated by the comparative *C*_T_ method (ΔΔ*C*_T_).

### 4.3. Western Blot Analysis

Ocular tissues, homogenized with the Precellys 24 homogenizer and lysing kit (Bertin; Frankfurt, Germany), and cultured cells were extracted with RIPA buffer (50 mM Tris-HCl, pH 8.0, 150 mM NaCl, 1% NP-40, 0.5% DOC, 0.1% SDS). Protein concentrations were determined with the Micro-BCA protein assay kit (Fisher Scientific; Schwerte, Germany). Proteins were separated by 4–15% SDS-polyacrylamide gel electrophoresis under reducing conditions (6% DTT) and transferred onto nitrocellulose membranes with the Trans-Blot Turbo transfer system (Bio-Rad Laboratories, München, Germany). Membranes were blocked with SuperBlock T20 (Fisher Scientific; Schwerte, Germany) for 30 min and incubated for 1 h at room temperature with antibodies diluted in phosphate-balanced saline (PBS) containing 0.1% Tween 20 (PBST) and 10% SuperBlock T20. Antibody source, species, and conditions are given in Appendix A. Equal loading was verified with anti-human ß-actin antibody in PBST/10% SuperBlock T20. In negative control experiments, the primary antibody was replaced by PBST. Immunodetection was performed with a horseradish peroxidase-conjugated goat anti-mouse (Biolegend; Amsterdam, The Netherlands) or goat anti-rabbit (Cell Signaling, Leiden, The Netherlands) secondary antibodies diluted 1:10,000 in PBST/10% SuperBlock T20 and the Super Signal West Femto ECL kit (Fisher Scientific; Schwerte, Germany). Band intensity was analyzed with the LAS-3000 chemiluminescence detection system and software (Multi Gauge V3.0) (Fujifilm; Düsseldorf, Germany).

### 4.4. Immunohisto- and Immunocytochemistry

Ocular tissue samples were embedded in optimal cutting temperature (OCT) compound and snap frozen in isopentane-cooled liquid nitrogen. Cryosections of 4 μm thickness were fixed in 4% paraformaldehyde for 10 min, washed in PBS, and permeabilized using 0.1% Triton X-100 in PBS for 10 min. After blocking with 10% normal goat serum, sections were incubated overnight at 4 °C in primary antibodies (Appendix A) diluted in PBS. Antibody binding was detected by Alexa Fluor^®^ 488- or 555-conjugated secondary antibodies (Life Technologies; Carlsbad, CA, USA). Nuclear counterstaining was achieved using 4,6′-diamino-2-phenylindole (DAPI; Sigma-Aldrich; St. Louis, MO, USA). Immunolabeled slides and cell cultures were washed and coverslipped with Vectashield mounting medium (Vector Laboratories; Burlingame, CA, USA) prior to evaluation on a fluorescence microscope (BX51, Olympus; Hamburg, Germany). In negative control experiments, the primary antibodies were replaced by equimolar concentrations of isotype-specific mouse and rabbit immunoglobulins or irrelevant isotypic primary antibodies.

For immunocytochemical analysis of extracellular matrix production, human Tenon’s capsule fibroblasts were seeded at a density of 100,000 cells/well into 4 well-chamber slides (Sarstedt; Nümbrecht, Germany), cultured for 7 days, and stained with antibodies against various matrix components (Appendix A). The fluorescence intensity of 20 images per well, captured at a magnification of 200×, was measured using the ZEN blue imaging software (Carl Zeiss Microscopy; Oberkochen, Germany).

### 4.5. Electrophoresis Mobility Shift Assays

Electrophoretic mobility shift assays (EMSA) were performed as described previously [12]. Briefly, 31 and 26 bp synthetic oligonucleotides (TCGAGGGTAGGGTTCACCGAAAGTTCACTCG and AGCTTCAGGTCAGAGGTCAGAGAGCT) containing RAR and RXR consensus motifs (underlined) were 5′-labeled with biotin. EMSA probes were applied as double strands after paired annealing at 95 °C for 5 min, followed by cooling down to room temperature for several hours. Nuclear extracts from primary human Tenon’s capsule fibroblasts obtained from PEX and control patients were generated by using a nuclear extraction kit (NE-PER Nuclear and Cytoplasmic Extraction reagents; Fisher Scientific; Schwerte, Germany). Each 20 µL binding reaction included salmon sperm DNA (final concentration: 2.5 ng/µL) as nonspecific competitor DNA. For competition experiments, a 200-fold molar excess of unlabeled oligonucleotides was included in the pre-incubation mixture. The resulting complexes were resolved on 6% native polyacrylamide gels. EMSA was performed using the Lightshift Chemiluminescent EMSA Kit (Fisher Scientific; Schwerte, Germany). Gel shifts were quantitatively analyzed with the LAS-3000 chemiluminescence detection system and software (Multi Gauge V3.0; Fujifilm; Düsseldorf, Germany).

### 4.6. ELISA

Retinol-binding protein 3 (RBP3) was analyzed in aqueous humor samples (diluted 1:7.5) of patients with PEX syndrome (n = 28) and cataract as control (n = 18) as well as serum samples (diluted 1:5) of patients with PEX syndrome (n = 12) and controls (n = 12) using an RBP3 Human ELISA Kit (Abnova; Taipeh, Taiwan) with a sensitivity of <0.094 pg/mL according to the manufacturer’s instructions. Retinol-binding protein 4 (RBP4) was analyzed in aqueous humor samples (diluted 1:500) of patients with PEX syndrome (n = 12) and cataract (n = 12) as well as serum samples (diluted 1: 400.000) of PEX patients (n = 10) and controls (n = 12) using a Human RBP4 ELISA Kit (Abcam; Cambridge, UK) with a sensitivity of 2.6 pg/mL. Transthyretin (prealbumin) was analyzed in aqueous humor samples (diluted 1:3000) of patients with PEX syndrome (n = 11) and cataract (n = 16) as well as serum samples (diluted 1: 200.000) of PEX patients (n = 11) and controls (n = 12) using a Human PreAlbumin ELISA Kit (Abcam; Cambridge, UK) with a sensitivity of 55 pg/mL. Samples were analyzed as single measurements. Total aqueous protein concentrations were determined using the Bradford protein assay kit (Bio-Rad; Munich, Germany) with bovine serum albumin as a standard.

### 4.7. Vitamin A Measurement

Serum vitamin A levels were measured in 34 patients with PEX syndrome and 34 patients with FECD. Peripheral venous blood was collected, protected from light, centrifuged at 3000 rpm for 10 min, and the concentration of vitamin A in serum samples was detected by high-performance liquid chromatography according to the standard method.

### 4.8. Cell Culture

Tenon’s capsule biopsies were obtained from four patients (mean age, 7.5 ± 1.2 years; 2 male, 2 female) during strabismus surgery, which were used for transfection experiments, and from four patients (mean age, 63.8 ± 4.3 years; 2 male, 2 female) during cataract surgery, which were used for stimulation experiments. Primary human Tenon’s capsule fibroblast (hTCF) cultures were established as previously described [62] and maintained in Dulbecco’s modified Eagle’s medium (DMEM/Ham’s F12; Fisher Scientific; Schwerte, Germany) containing 15% (*v*/*v*) fetal bovine serum (FBS) and 1% antibiotic-antimycotic solution (PSA). Primary human trabecular meshwork cells (hTMC) from four different donors (fetal to 24 years; 2 male, 2 female) were obtained from Provitro (Berlin, Germany) and grown in DMEM supplemented with 10% FBS and 1% PSA. The immortalized human nonpigmented ciliary epithelial (NPE) cell line ODM-2 [63] was used at passage 16 and maintained in DMEM containing 10% FBS and 50 μg/mL gentamicin. Primary iris pigment epithelial (IPE) cells from three different donors (mean age, 22.0 ± 0.8 years; 2 male, 1 female) were obtained from Provitro (Berlin, Germany) and grown in DMEM/F12 supplemented with 10% FBS and 1% PSA. The immortalized lens epithelial (LEP) cell line B-3 was obtained from LGC (Wesel, Germany), used at passage 4, and grown in MEM supplemented with 20% FBS, 1% non-essential amino acids, and 1% PSA. All primary cells were used in passage 4. Cells were maintained at 37 °C in a 95% humidified 5% CO_2_ atmosphere.

Cells were stimulated with the following RA pathway inhibitors (Tocris Bioscience; Bristol, UK) for 48 h under serum-reduced conditions (0.5% FBS): aldehyde dehydrogenase inhibitors disulfiram (1 and 5 µM) and liarozole (10 and 100 nM), pan-RAR antagonists AGN 193109 (1 and 5 µM) and BMS 493 (1 and 5 µM), and pan-RXR antagonists HX 531 (10 and 20 nM) and UVI 3003 (10 and 20 µM). For RA pathway activation, cells were stimulated with natural retinoids all-*trans* RA (ATRA) and 9-*cis* RA (Sigma-Aldrich; St. Louis, MO, USA) at concentrations of 0.5–10 µM or with the following synthetic small molecule retinoids (Tocris Bioscience; Bristol, UK) at concentrations of 0.01–10 µM for 48 h under serum-reduced conditions: the specific RARα agonist AM 80, the specific RARβ agonist AC 261066, the specific RARγ agonist CD 1530, the pan-RXR agonist LGD 1069 (bexarotene), and the specific RXRα agonist CD 3254. In additional experiments, cells were stimulated with 5 ng/mL TGF-β1 (R&D Systems; Minneapolis, MN, USA) for 48 h without or with 2 µM ATRA or the TGF-β1 inhibitor SB 431542 (Sigma-Aldrich; St. Louis, MO, USA).

Three-dimensional collagen lattices were prepared using the 3D Collagen Cell Culture Kit (Merck; Darmstadt, Germany). Briefly, 1 × 10^6^ cells/mL (hTCF) were suspended in a cooled collagen gel solution. 500 μL of this suspension was aliquoted into each well of a 24-well culture plate and allowed to polymerize at 37 °C for 60 min. Collagen gels were covered with 500 µL of serum-reduced DMEM containing 5 ng/mL TGF-β1 or 5 ng/mL TGF-β1 plus 2 µM ATRA and cultured for 10 days. Serum-reduced medium without any compound served as control. Media were changed daily.

### 4.9. siRNA Silencing

hTCF and hTMC were transiently transfected with 150 pmol per 1 × 10^6^ cells specific siRNA (ON-TARGETplus SMARTpool; GE Healthcare Dharmacon; Freiburg, Germany) against *RARA* or *RXRA* by electroporation using the Nucleofector II transfection device and the Amaxa Basic Fibroblasts Nucleofector Kit (Lonza; Köln, Germany) with the nucleofector program U-023. Transfections with scrambled siRNA (ON-TARGETplus Non-targeting pool) served as controls. Transfected cells were seeded into 6-well plates and harvested at 48 h post-transfection for real-time PCR analysis.

### 4.10. Transmission Electron Microscopy

After 10 days of incubation, the collagen gels were fixed in 2.5% glutaraldehyde in a 0.1M phosphate buffer, postfixed in 2% buffered osmium tetroxide, dehydrated in graded alcohol concentrations, and embedded in epoxy resin according to standard protocols. Ultrathin sections were stained with uranyl acetate-lead citrate and examined with a transmission electron microscope (EM 906E; Carl Zeiss Microscopy, Oberkochen, Germany).

### 4.11. Statistical Analysis

Statistical analyses were performed using the GraphPad InStat statistical package for Windows (Version 8.3.0; GraphPad Software Inc., La Jolla, CA, USA). Data are expressed as mean ± standard deviation from individual experiments. Gaussian distribution was tested using Anderson–Darling or Kolmogorov–Smirnov tests, depending on case numbers. Group comparisons and correlation analyses were performed using an unpaired two-tailed *t*-test or a Mann–Whitney *U* test and Pearson or Spearman tests, respectively, depending on the presence of parametric or nonparametric data sets. A *p*-value of <0.05 was considered statistically significant.

## Figures and Tables

**Figure 1 ijms-23-05977-f001:**
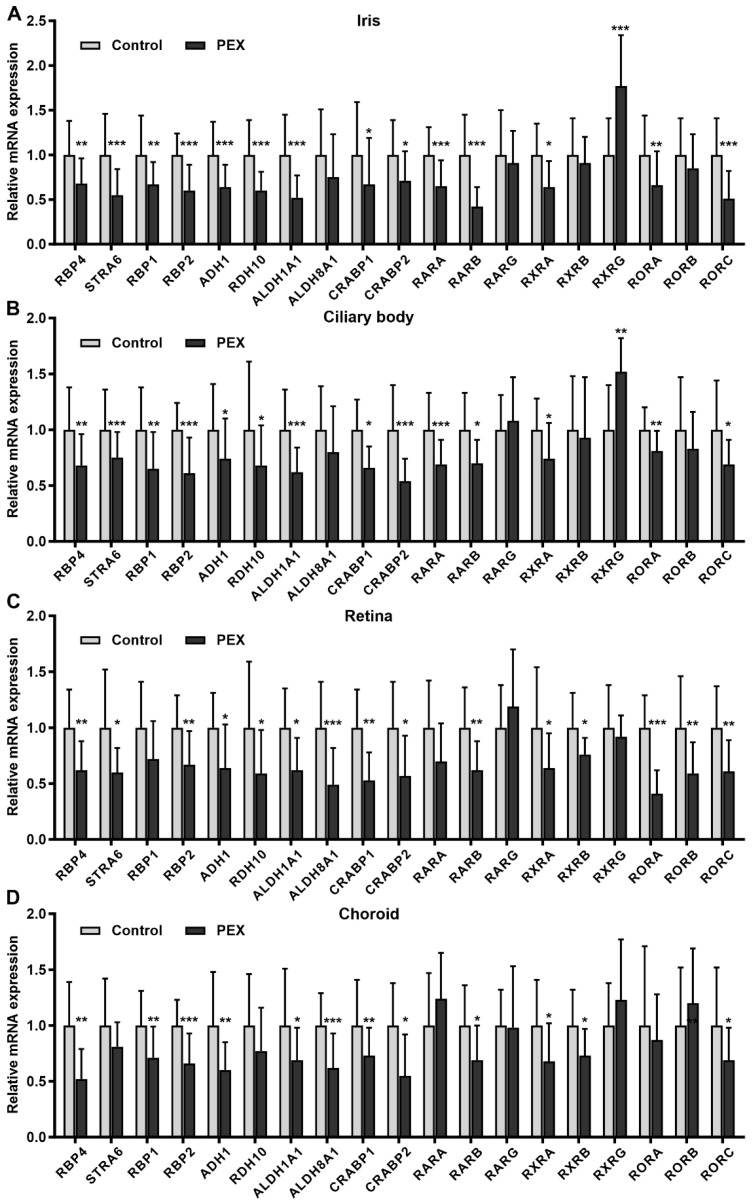
Comparative analysis of retinoic acid pathway components in ocular tissues of PEX and control eyes. Quantitative real-time PCR analysis of (**A**) iris, (**B**) ciliary body, (**C**) retina, and (**D**) choroid tissue samples from unaffected donors (Control, n = 23) and PEX patients (n = 24) showing relative mRNA expression levels of *RBP4* (retinol binding protein 4, plasma), *STRA6* (signaling receptor and transporter of retinol STRA6), *RBP1*, *RBP2* (retinol binding protein, cellular), *ADH* (alcohol dehydrogenase), *RDH* (retinol dehydrogenase), *ALDH* (aldehyde dehydrogenase), *CRABP* (cellular retinoic acid binding protein), *RAR* (retinoic acid receptor), *RXR* (retinoid X receptor), and *ROR* (RAR related orphan receptor). Data are normalized to *GAPDH* (glyceraldehyde-3-phosphate dehydrogenase) and *HPRT1* (hypoxanthine phosphoribosyltransferase 1) and expressed as means ±  SD relative to controls set to 1 (* *p* < 0.05, ** *p* < 0.01, *** *p* < 0.001; unpaired *t*-test).

**Figure 2 ijms-23-05977-f002:**
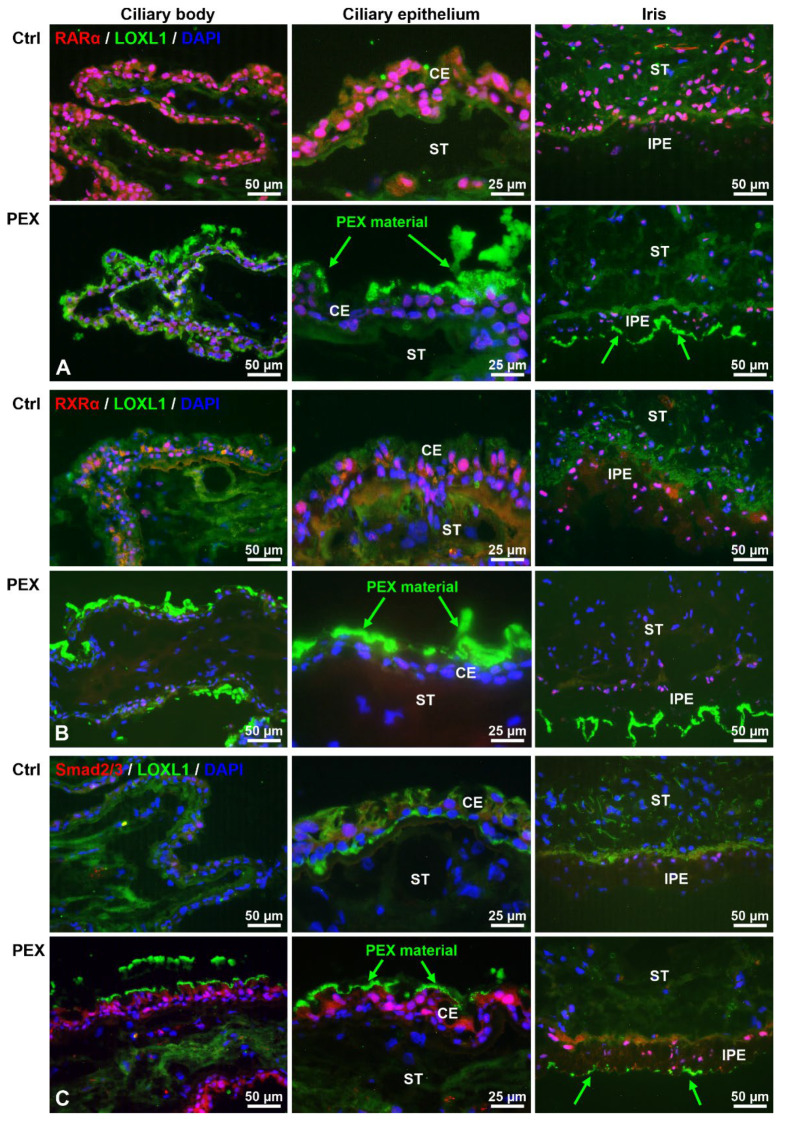
Immunohistochemical localization of RARα, RXRα, LOXL1, and Smad 2/3 in ocular tissues of normal human control (Ctrl) and PEX eyes. Double labeling of RARα (**A**), RXRα (**B**), and Smad2/3 (**C**) (red fluorescence) in cell nuclei and LOXL1 (green fluorescence) in PEX material accumulations (arrows) on ocular surfaces; nuclear counterstaining with 4′,6-diamidino-2-phenylindole (DAPI, blue fluorescence). CE, ciliary epithelium; IPE, iris pigment epithelium; ST, stroma.

**Figure 3 ijms-23-05977-f003:**
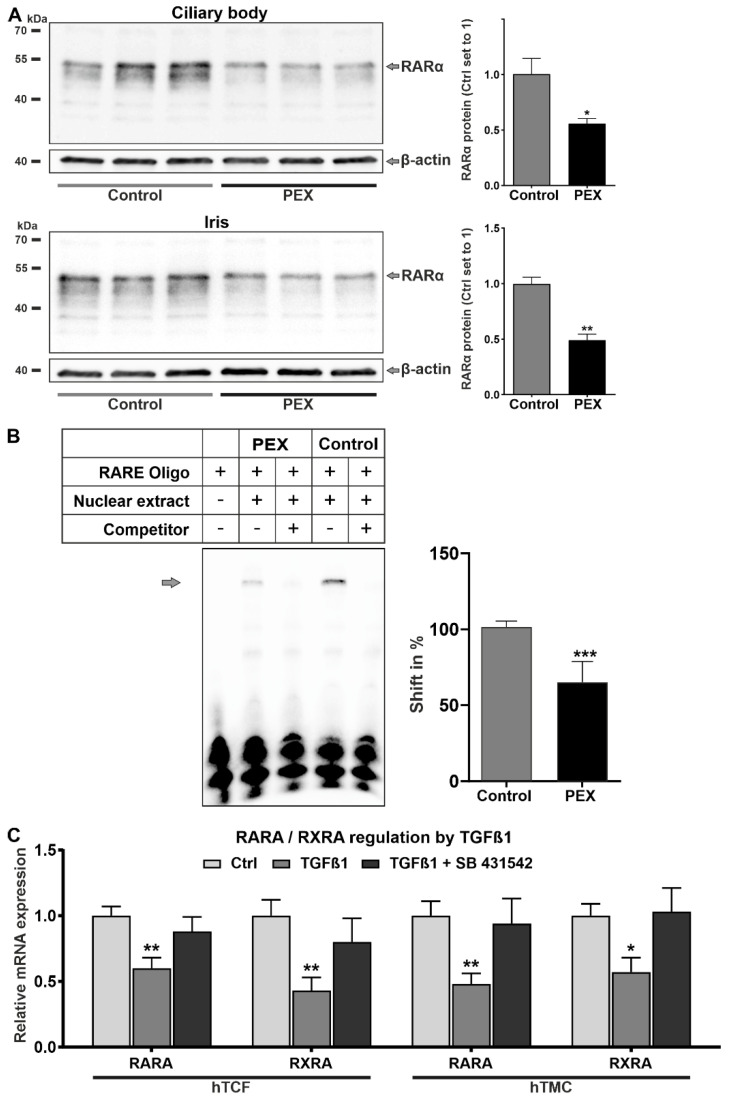
Expression, activity, and regulation of nuclear receptors RARα and RXRα in ocular tissues of PEX and control eyes. (**A**) Western blot analysis of RARα expression in the ciliary body and iris tissue from PEX patients compared to normal donors (n = 3). Protein expression is normalized to the house-keeping gene β-actin and is expressed relative to expression in controls (set to 1). (**B**) Electrophoretic mobility shift assay using oligonucleotides containing retinoic acid response element (RARE) consensus binding sequences and nuclear extracts (4 µg) from human Tenon’s capsule fibroblasts derived from PEX and control eyes. Unlabeled oligonucleotides were used as competitors. Quantitative analysis of the protein-DNA complexes shows mean values ± SD (n = 5) relative to the control set to 100%. (**C**) Real-time PCR analysis of *RARA* and *RXRA* regulation by TGF-β1 without and with TGF-β1 inhibitor SB 431542 in cultured human Tenon’s capsule fibroblasts (hTCF, n=3) and human trabecular meshwork cells (hTMC, n = 3) (* *p* < 0.05, ** *p* < 0.01, *** *p* < 0.001; unpaired *t*-test).

**Figure 4 ijms-23-05977-f004:**
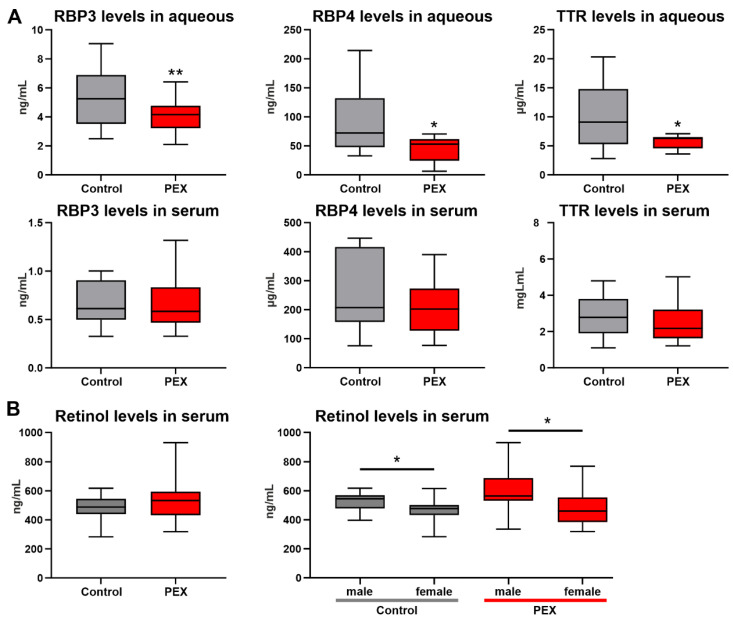
Levels of retinol and retinol carrier proteins in aqueous humor and serum of PEX and control patients. (**A**) ELISA analysis of levels of RBP3 (retinol-binding protein 3), RBP4 (retinol-binding protein 4), and TTR (transthyretin) in aqueous humor (n ≥ 15) and serum samples (n = 12) of PEX and control patients. (**B**) HPLC analysis of serum retinol levels in PEX and control patients (n = 34). Data are presented as box-and-whisker plots displaying the median (line), lower and upper quartiles (boxes), and minimum-maximum (whiskers). Asterisks indicate significant differences between groups (* *p* < 0.05, ** *p* < 0.01; unpaired *t*-test).

**Figure 5 ijms-23-05977-f005:**
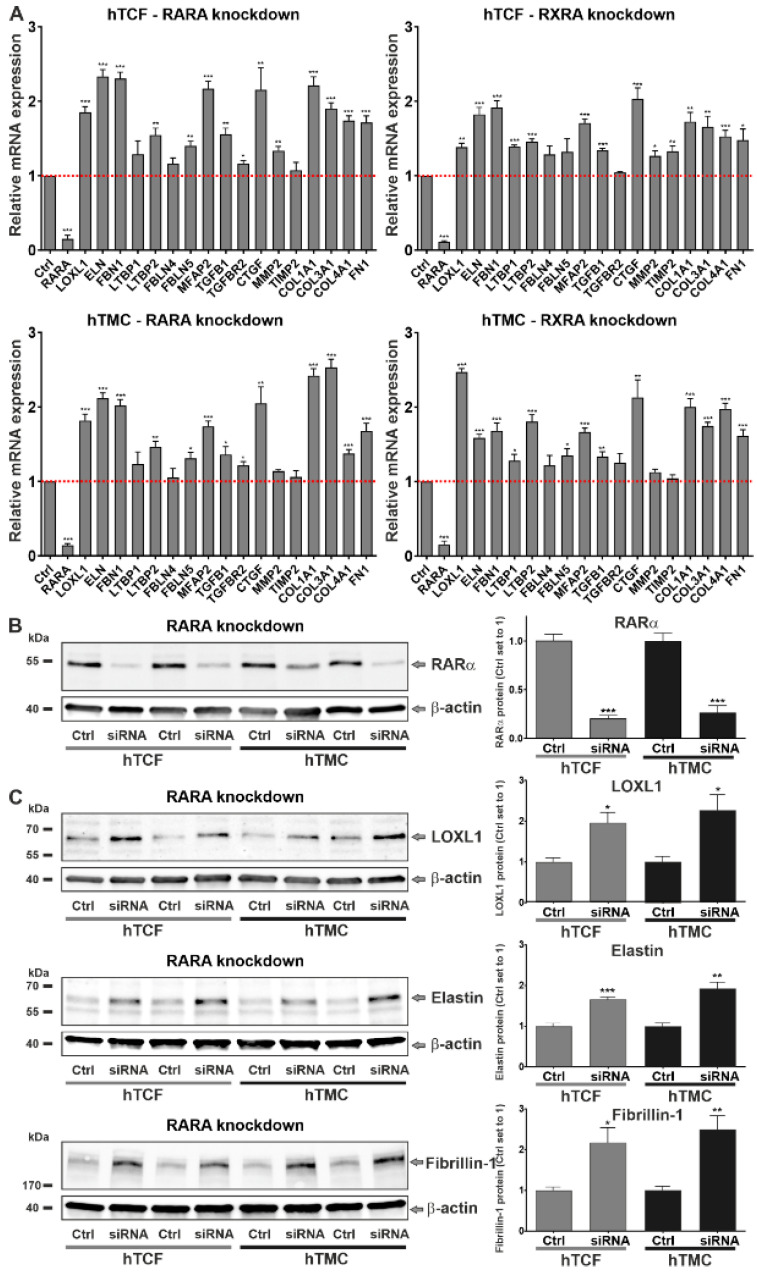
Effects of *RARA* (retinoic acid receptor alpha) and *RXRA* (retinoid X receptor alpha) knockdown on matrix synthesis in PEX-relevant cell types. (**A**) Quantitative real-time PCR analysis of *RARA*, *RXRA*, *LOXL1* (lysyl oxidase-like 1)*, ELN* (elastin), *FBN1* (fibrillin-1), *LTBP1* (latent transforming growth factor beta binding protein 1), *LTBP2*, *FBLN4* (fibulin-4), *FBLN5*, *MFAP2* (microfibril associated protein 2), *TGFB1* (transforming growth factor beta 1), *TGFBR2* (transforming growth factor beta receptor 2), *CTGF* (connective tissue growth factor), *MMP2* (matrix metalloproteinase 2), *TIMP2* (tissue inhibitor of metalloproteinases 2), *COL1A1* (collagen type 1 alpha 1), *COL3A1*, *COL4A1*, and *FN1* (fibronectin-1) mRNA in human Tenon’s capsule fibroblasts (hTCF) (n = 4) and trabecular meshwork cells (hTMC) (n = 4) transfected with *RARA*- or *RXRA*-specific siRNA or scrambled control siRNA. Expression levels were normalized to *GAPDH* and *HPRT1* and expressed as means ± SD relative to controls set to 1 (red dashed line). (**B**,**C**) Western blot analysis of RARα (**B**) as well as LOXL1, elastin, and fibrillin-1 (**C**) protein after *RARA* siRNA-mediated gene silencing in hTCF and hTMC (n = 4) compared to cells transfected with scrambled non-targeting siRNA (Ctrl). Protein expression is normalized to the house-keeping gene β-actin and is expressed relative to expression in controls (set to 1); (* *p* < 0.05, ** *p* < 0.01, *** *p* < 0.001; unpaired *t*-test).

**Figure 6 ijms-23-05977-f006:**
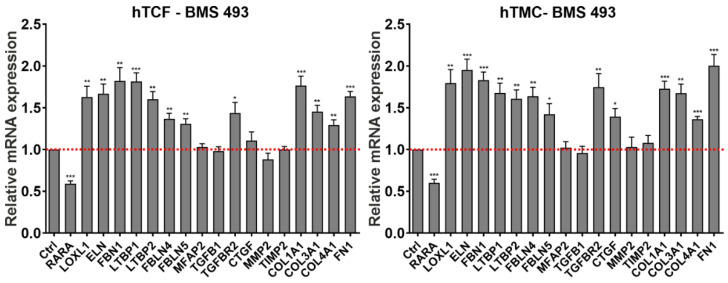
Effects of the RAR antagonist BMS 493 on matrix synthesis in PEX-relevant cell types. Quantitative real-time PCR analysis of *RARA* (retinoic acid receptor alpha), *LOXL1* (lysyl oxidase-like 1)*, ELN* (elastin), *FBN1* (fibrillin-1), *LTBP1* (latent transforming growth factor beta binding protein 1), *LTBP2*, *FBLN4* (fibulin-4), *FBLN5*, *MFAP2* (microfibril associated protein 2), *TGFB1* (transforming growth factor beta 1), *TGFBR2* (transforming growth factor beta receptor 2), *CTGF* (connective tissue growth factor), *MMP2* (matrix metalloproteinase 2), *TIMP2* (tissue inhibitor of metalloproteinases 2), *COL1A1* (collagen type 1 alpha 1), *COL3A1*, *COL4A1*, and *FN1* (fibronectin-1) mRNA in human Tenon’s capsule fibroblasts (hTCF) (n = 4) and trabecular meshwork cells (hTMC) (n = 4) treated with the pan-RAR antagonist BMS 493 at a concentration of 5 µM. Expression levels were normalized to *GAPDH* and *HPRT1* and expressed as means ± SD relative to untreated controls set to 1 (red dashed line); (* *p* < 0.05, ** *p* < 0.01, *** *p* < 0.001; unpaired *t*-test).

**Figure 7 ijms-23-05977-f007:**
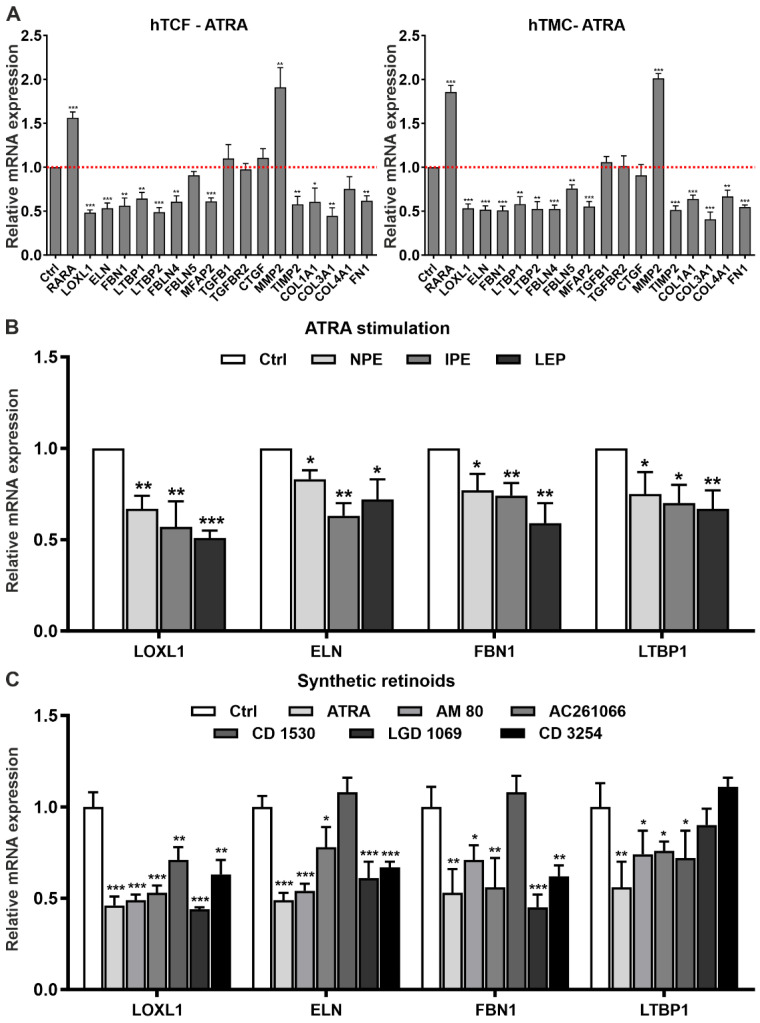
Effects of retinoic acid signaling activation on matrix synthesis in PEX-relevant cell types. (**A**) Quantitative real-time PCR analysis of *RARA* (retinoic acid receptor alpha), *LOXL1* (lysyl oxidase-like 1)*, ELN* (elastin), *FBN1* (fibrillin-1), *LTBP1* (latent transforming growth factor beta binding protein 1), *LTBP2*, *FBLN4* (fibulin-4), *FBLN5*, *MFAP2* (microfibril associated protein 2), *TGFB1* (transforming growth factor beta 1), *TGFBR2* (transforming growth factor beta receptor 2), *CTGF* (connective tissue growth factor), *MMP2* (matrix metalloproteinase 2), *TIMP2* (tissue inhibitor of metalloproteinases 2), *COL1A1* (collagen type 1 alpha 1), *COL3A1*, *COL4A1*, and *FN1* (fibronectin-1) mRNA in human Tenon’s capsule fibroblasts (hTCF) (n = 4) and trabecular meshwork cells (hTMC) (n = 4) treated with 2 µM all-*trans* retinoic acid (ATRA). (**B**) Relative mRNA expression levels of *LOXL1*, *ELN*, *FBN1*, and *LTBP1* in human non-pigmented ciliary epithelial (NPE), iris pigment epithelial (IPE) and lens epithelial (LEP) cells treated with 2 µM ATRA. (**C**) Relative mRNA expression levels of *LOXL1*, *ELN*, *FBN1*, and *LTBP1* in hTCF treated with 2 µM ATRA or 10 µM of synthetic agonists selective for RARα (AM 80), RARβ (AC 261066), RARγ (CD 1530), RXRα (CD 3254) and pan-RXR (LGD 1069). Expression levels were normalized to *GAPDH* and *HPRT1* and expressed as means ± SD relative to untreated controls set to 1; (* *p* < 0.05, ** *p* < 0.01, *** *p* < 0.001; unpaired *t*-test).

**Figure 8 ijms-23-05977-f008:**
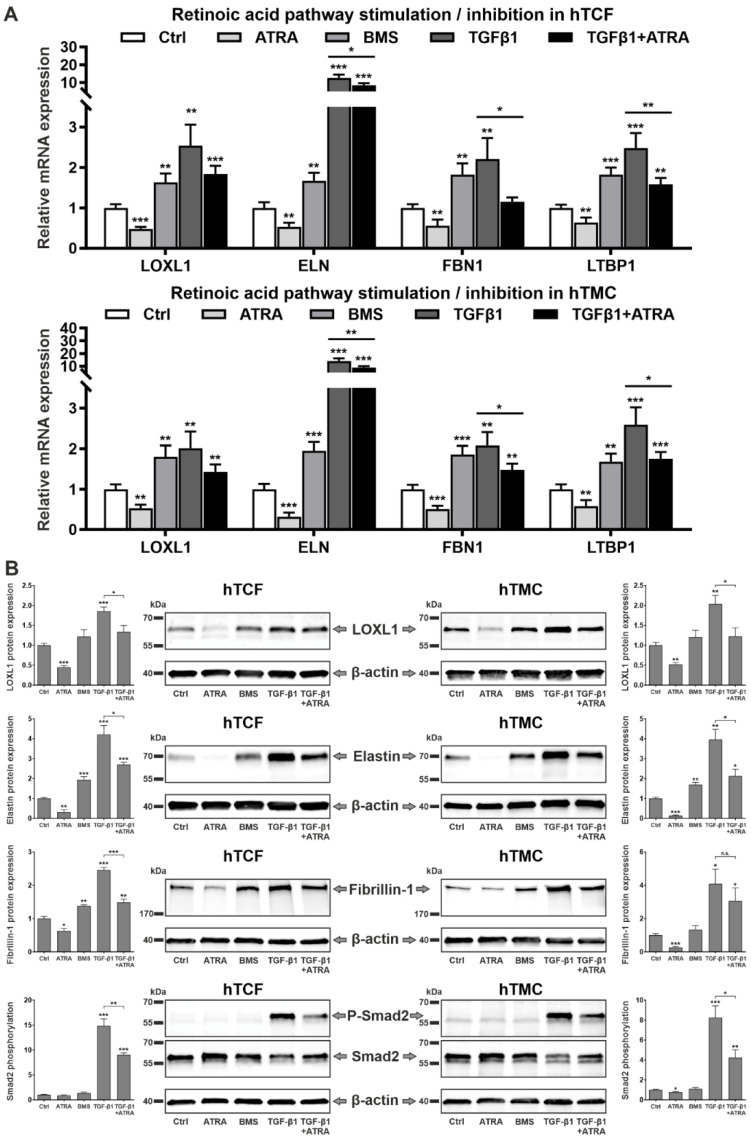
Effects of retinoic acid signaling activation and inhibition on TGF-β1-induced matrix synthesis in PEX-relevant cell types. (**A**) Quantitative real-time PCR analysis of *LOXL1* (lysyl oxidase-like 1)*, ELN* (elastin), *FBN1* (fibrillin-1), and *LTBP1* (latent transforming growth factor beta binding protein 1) mRNA in human Tenon’s capsule fibroblasts (hTCF) (n = 4) and trabecular meshwork cells (hTMC) (n = 4) treated with 2 µM all-*trans* retinoic acid (ATRA), 5 µM BMS 493 (BMS, pan-RAR antagonist), 5 ng/mL transforming growth factor-ß1 (TGF-β1), or 5 ng/mL TGF-β1 together with 2 µM ATRA. (**B**) Western blot analysis of LOXL1, elastin, and fibrillin-1 protein expression as well as total and phosphorylated Smad2 in hTCF and hTMC without stimulation (control, Ctrl) or in response to 2 µM ATRA, 5 µM BMS 493, 5 ng/mL TGF-β1, or 5 ng/mL TGF-β1 together with 2 µM ATRA (n = 4). Expression levels were normalized to *GAPDH* and *HPRT1* (PCR) and β-actin (Western blot), respectively, and expressed as means ± SD relative to untreated controls set to 1 (* *p* < 0.05, ** *p* < 0.01, *** *p* < 0.001; unpaired *t*-test).

**Figure 9 ijms-23-05977-f009:**
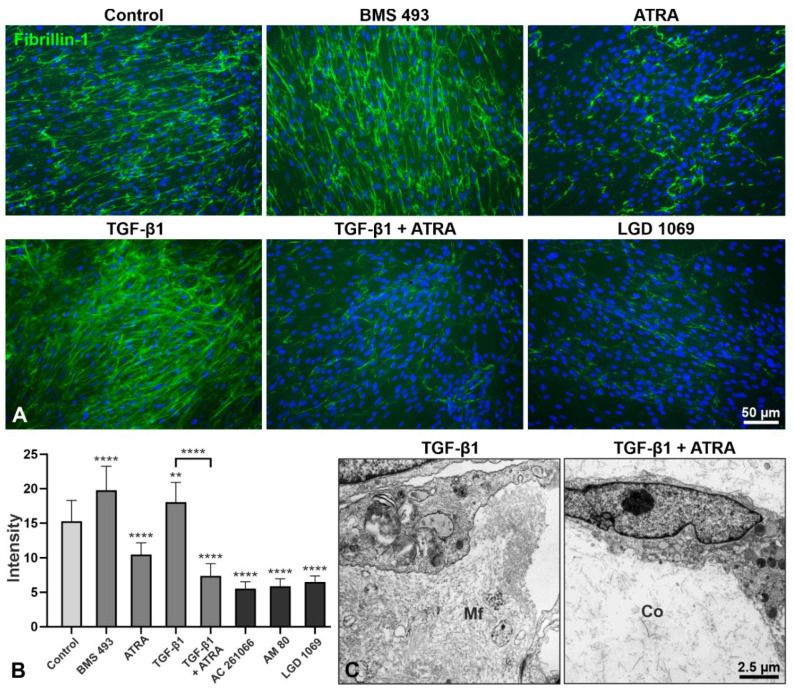
Immunofluorescence analysis of microfibrillar network assembly in response to retinoic acid pathway inhibition and activation. (**A**) Immunolabeling for fibrillin-1 (green fluorescence) in cultures of human Tenon’s capsule fibroblasts (hTCF) treated with 5 µM of BMS 493 (pan-RAR antagonist), 2 µM all-*trans* retinoic acid (ATRA), 5 ng/mL transforming growth factor-β1 (TGF-β1), 5 ng/mL TGF-β1 together with 2 µM ATRA, or 10 µM of LGD 1069 (pan-RXR agonist) for 7 days; untreated cells served as control; nuclear counterstaining with 4′,6-diamidino-2-phenylindole (DAPI, blue fluorescence). (**B**) Quantitative analysis of fluorescence intensity of fibrillin-1 staining across whole images (n = 20 per group). Data are expressed as means ± SD (** *p* < 0.01, **** *p* < 0.0001; unpaired *t*-test). (**C**) Transmission electron micrographs showing the production of microfibrils (Mf) by hTCF cultivated in 3D collagen (Co) gels following treatment with 5 ng/mL TGF-β1 for 10 days. The addition of 2 µM ATRA to TGF-β1 inhibited the deposition of microfibrils within the collagen gels.

**Figure 10 ijms-23-05977-f010:**
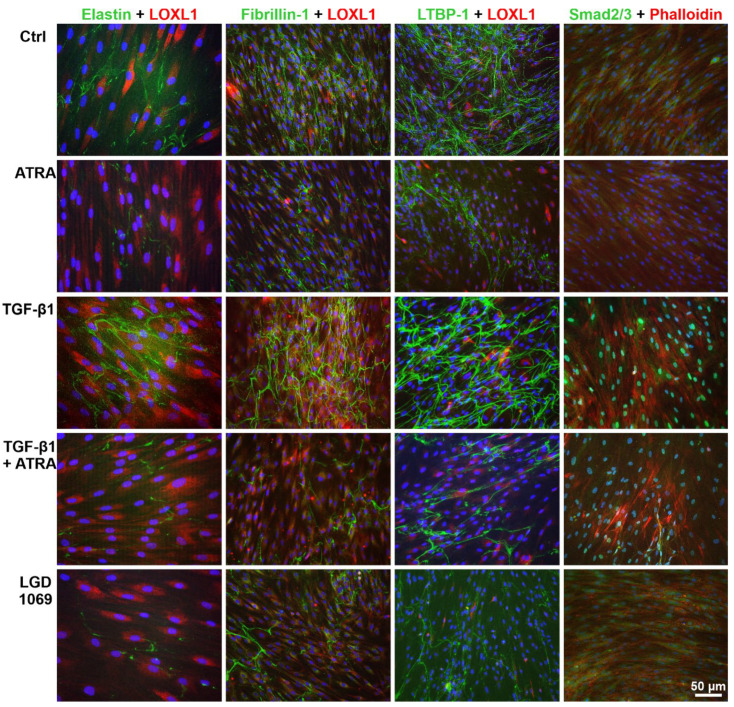
Immunofluorescence analysis of microfibrillar network composition in response to retinoic acid pathway activation. Immunofluorescent double labeling for elastin, fibrillin-1, LTBP-1, or Smad2/3 (green fluorescence) and LOXL1 or Alexa Fluor 555-conjugated phalloidin (red fluorescence) in cultures of human Tenon’s capsule fibroblasts treated with 2 µM all-trans retinoic acid (ATRA), 5 ng/mL transforming growth factor-β1 (TGF-β1), 5 ng/mL TGF-β1 plus 2 µM ATRA, or 10 µM of LGD 1069 (pan-RXR agonist) for 7 days; untreated cells served as control (Ctrl); nuclear counterstaining with 4′,6-diamidino-2-phenylindole (DAPI, blue fluorescence).

## Data Availability

The datasets generated in this study are available from the corresponding author on request.

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
