# Peer review of "Dysregulated Retinoic Acid Signaling in the Pathogenesis of Pseudoexfoliation Syndrome"

_ijms, 2022, doi:10.3390/ijms23115977_

Round 1

Reviewer 1 Report

In this manuscript, Zenkel et al. have investigated a causative role of the retinoic acid pathway in the pathogenesis of pseudoexfoliation syndrome. Authors have established a connection between dysregulated expression of proteins involved in RA pathway to that of TGF-ß1/Smad signalling in eye tissues. Most importantly, they have shown that PEX-associated fibrosis can be suppressed by activation of RA pathway through retinoids. Overall data presented in the manuscript justifies the conclusions made. Detailed study design and presentation of the data in the manuscript is of high quality and was pleasant to read. The reviewer has following few minor suggestions.

Authors should consider using multiple testing correction (such as Bonferroni adjustment) while analyzing the statistical significance for differential expression of multiple genes, especially when there is a fold difference of ~0.5. 

Abbreviations should be expanded at their first occurrence such as for hTCF (line 192) and htmc (line 198).

Lines 211-215 should be incorporated as figure legend instead of the with main text.

Amount of nuclear extract used for binding assays (Fig 3B)?

Author Response

Authors should consider using multiple testing correction (such as Bonferroni adjustment) while analyzing the statistical significance for differential expression of multiple genes, especially when there is a fold difference of ~0.5.

Reply:

Group comparisons were performed using an unpaired two-tailed t test or Mann-Whitney U test for parametric or non-parametric data sets, respectively, and were separately performed as single column analyses, comparing expression levels of each gene in “diseased and normal control state” or “treated and untreated control conditions”. No multiple tests, such as one-way ANOVA requiring Bonferroni correction, were applied. We therefore hope that the analysis of statistical significance, as performed in this study, would be appropriate. 

Abbreviations should be expanded at their first occurrence such as for hTCF (line 192) and htmc (line 198).

Reply:

Thank you, this has been corrected.

Lines 211-215 should be incorporated as figure legend instead of the with main text.

Reply: 

We are sorry for this trouble, which arose during automatic re-formatting of the submitted word file by the journal. We hope this could be settled in the revised version.

Amount of nuclear extract used for binding assays (Fig 3B)?

Reply:

The amount of nuclear extract used was 4 µg. We included this information in the legend to Fig. 3B and changed the number of binding assays performed to n=5. 

Reviewer 2 Report

Pseudoexfoliation syndrome (PEX) is the commonest recognizable cause of open-angle glaucoma. The recent identification of PEX-associated gene variants uncovered the vitamin A metabolic pathway as a factor influencing risk of disease. Authors analyzed the role of the retinoic acid signaling pathway in PEX-associated matrix metabolism and evaluated its targeting as a potential candidate for an anti‑fibrotic intervention. Their findings indicate that deficient retinoic acid signaling in conjunction with hyperactivated TGF-ß1/Smad signaling is a driver of PEX-associated fibrosis, and that restoration of retinoic acid signaling may be a promising strategy for anti-fibrotic intervention in patients with PEX syndrome and glaucoma. This manuscript reviews 63 articles and provides a complex survey of current literature dealing with this topic. The topic of this manuscript is up to date, interesting and well suited for the International Journal of Molecular Sciences. The manuscript is well written and divided into 4 parts, the text is clear and easy to read. For better understanding authors used 10 illustrations. This aids the reader's understanding. I suggest checking for some small spelling mistakes and grammar errors. Otherwise, I have no major concerns about this manuscript and I recommend it for publication.

Author Response

Reply:

Thank you very much for the appreciation of our work. We have tried to corect all spelling mistakes and grammar errors in the manuscript.